# Analysis of Intestinal Mucosa Integrity and *GLP-2* Gene Functions upon Porcine Epidemic Diarrhea Virus Infection in Pigs

**DOI:** 10.3390/ani11030644

**Published:** 2021-03-01

**Authors:** Yajing Zhou, Zhanshi Ren, Shuai Zhang, Haifei Wang, Shenglong Wu, Wenbin Bao

**Affiliations:** 1Key Laboratory for Animal Genetics, Breeding, Reproduction and Molecular Design, College of Animal Science and Technology, Yangzhou University, Yangzhou 225009, China; xxxcandygd@163.com (Y.Z.); Renzhanshi0422@163.com (Z.R.); shuai_zhang1990@163.com (S.Z.); wanghaiffei@126.com (H.W.); slwu@yzu.edu.cn (S.W.); 2Joint International Research Laboratory of Agriculture & Agri-Product Safety, The Ministry of Education of China, Yangzhou University, Yangzhou 225009, China

**Keywords:** pigs, PEDV, *GLP-2* gene, intestinal mucosa barrier, gene expression

## Abstract

**Simple Summary:**

Porcine epidemic diarrhea virus causes serious diarrhea in suckling piglets, resulting in huge economic losses in the pig industry. A better understanding of porcine epidemic diarrhea virus (PEDV) pathogenesis and identification of the responsive genes will contribute to controlling PEDV infection. Therefore, screening and identifying PEDV-resistance functional genes and then implementing disease resistance breeding have important scientific significance. This study explores the regulatory roles of the *GLP-2* gene in regulating PEDV infection and in repairing the intestinal mucosa damage associated with PEDV infection. The findings may contribute to the identification of resistant genes or genetic markers for disease resistance breeding in pigs.

**Abstract:**

Porcine epidemic diarrhea virus (PEDV) infects intestinal epithelial cells, destroys the intestinal mucosal barrier and then causes diarrhea in piglets. Glucagon-like peptide-2 (GLP-2) is a specific intestinal growth hormone that promotes the repair of damaged intestinal mucosa and improves the intestinal barrier. In this study, we investigated the functions of porcine *GLP-2* gene in regulating PEDV infection. The intestinal tissues with damaged intestinal structures caused by PEDV infection were first confirmed and collected. Expression analysis indicated that the *GLP-2* gene was expressed in the duodenum, jejunum and ileum tissues, and the mRNA level was significantly down-regulated in jejunum and ileum of piglets with damaged intestinal mucosa. Infection of PEDV to porcine small intestinal epithelial cells in vitro showed that *GLP-2* gene was significantly decreased, which was consistent with the expression pattern in intestinal tissues. In addition, we silenced the *GLP-2* gene by shRNA interfering and found that the copy numbers of PEDV were remarkably increased in the *GLP-2* gene silencing cells. Our findings suggest that the *GLP-2* gene was potentially involved in regulating PEDV infection and in maintaining the integrity of the intestinal mucosal barrier structure, which could contribute to our understanding of the mechanisms of PEDV pathogenesis and provide a theoretical basis for the identification and application of resistant genes in pig selective breeding for porcine epidemic diarrhea.

## 1. Introduction

Porcine epidemic diarrhea is a contagious disease caused by porcine epidemic diarrhea virus (PEDV), with clinical manifestations including severe diarrhea, vomiting, anorexia and dehydration. PEDV can infect pigs at all stages of growth and result in high mortality for suckling piglets, causing huge economic losses to the pig industry. PEDV, an invasive single-stranded positive-stranded RNA virus, belongs to the genus *Alphacoronavirus* in the family *Coronaviridae* [1]. The virus genome is about 28 kb in length, encoding two polyproteins (pp1a and pp1b), an accessory protein (open reading frame 3, ORF3) and four structural proteins (spike, S; envelope, E; membrane, M; and nucleocapsid, N) [2,3]. ORF3 protein causes the release of the virus by forming ion channels in the host [4]; S protein binds to the host surface receptor to regulate the replication of PEDV [5]; M protein interacts with S and N proteins, which plays an important role in the process of virus budding, assembly and replication [6,7]. PEDV mainly infects the intestinal villi cells and mesenteric lymph nodes of the pig small intestine. It destroys the structure of the intestine, damages the intestinal epithelial cells, causes nutrient absorption disorders and then causes diarrhea [8].

The intestine is an important part of the systemic immune system [9]. The intestinal mucosal barrier is a complex defense system composed of intestinal mucosal epithelial cells, intestinal mucosal immune system and intestinal microbes. The intestinal mucosal barrier protects the body by preventing pathogens, antigens and bacterial toxins from invading the mucosa and preventing the absorption of toxins [10]. It has been reported that as the most important component of the intestinal mucosal barrier, the intestinal mucosal epithelial cells play a regulatory role through tight junction proteins and mucins and are the first line of defense against gastrointestinal infections [11]. Tight junctions are an important connection method of the intestinal epithelium. The impaired intestinal barrier function of pigs is often related to the impaired tight junctions [12,13]. The intestinal mucosal immune system resists the invasion of pathogenic microorganisms through immunosuppression and protects important physiological functions of the body [14]. PEDV mainly invades the body through intestinal epithelial cells; therefore, intestinal mucosal epithelial cells and the intestinal mucosal immune system play an important role in defense against PEDV infection.

Glucagon-like peptide-2 (GLP-2) is a 33-amino-acid peptide, which is mainly composed of the post-translational processing of the glucagon pro-chain in the small intestinal endocrine L cells. It was produced in the small intestine along with GLP-1. The release of GLP-2 is mainly caused by direct contact with long-chain fatty acids in the terminal ileum. Studies have shown that GLP-2 is mediated by a specific G protein-coupled receptor, which is only found in the surrounding tissues of the intestine, such as L cells adjacent to intestinal crypts, enteric neuron and scattered populations of myofibroblasts [15,16,17]. As a specific intestinal growth hormone, GLP-2 inhibits cell apoptosis by promoting the proliferation and differentiation of intestinal cells [18]. It can also reduce the permeability of intestinal epithelial cells, promote the repair of damaged intestinal mucosa and improve the intestine tract barrier function [19,20]. Moran et al. [21] found that GLP-2 can promote the repair of intestinal mucosa by promoting the expression of tight junction proteins Zona Occluden-1 (ZO-1), Occludin and Claudine-1.

PEDV invades the body mainly through intestinal epithelial cells. The *GLP-2* gene plays a biological function in intestinal mucosal epithelial cells and the intestinal mucosal immune system. However, little is known about the regulatory effects of *GLP-2* gene on PEDV-induced intestinal injury. The aim of this study was to explore the effects of *GLP-2* gene expression on intestinal mucosal integrity and on the regulation of PEDV infections. Our findings could provide a theoretical basis for further research on the role and molecular mechanism of *GLP-2* gene in resisting porcine epidemic diarrhea.

## 2. Materials and Methods

### 2.1. Ethics Statement

The animal study proposal was approved by the Institutional Animal Care and Use Committee (IACUC) of the Yangzhou University Animal Experiments Ethics Committee (permit number: SYXK (Su) IACUC 2012-0029). All experimental methods were conducted in accordance with the related guidelines and regulations.

### 2.2. Experimental Animal

The experimental pigs were selected from a large-scale pig farm in Jiangsu Province. Six 7-day-old ternary hybrid piglets that were confirmed to be infected with PEDV by qPCR and six healthy piglets with the same birth weight through half-sib selection were used [22]. Two groups of piglets were slaughtered, and then tissue samples of duodenum, jejunum and ileum were collected, stored in liquid nitrogen on-site and transferred to −70 °C refrigerator for storage for later use.

### 2.3. Paraffin Section Preparation and Hematoxylin and Eosin (H&E) Staining

We separated the middle part of the duodenum, the jejunum and the ileum by about 3 cm. Then these tissue samples were rinsed with PBS and fixed in 4% paraformaldehyde solution for 24 h. They were sliced after dehydration, transparency, wax immersion and embedding. They were put in 45 °C and then took out the slices and baked slices at 60 °C for 1 h. After H&E staining, the morphology of each intestine segment was observed with a microscope (Leica, Heidelberg, Germany).

### 2.4. Primer Design and Synthesis

The porcine *GLP-2* (NM_214324.1) gene and PEDV M (AF017079.1) gene sequences published in the GenBank database, Premier 5.0 software, was used to design qPCR primers, with *GAPDH* and *ACTB* genes as internal control. Primers were synthesized by Sangon Biotech Co., Ltd. (Shanghai, China). Primer information is shown in Table 1.

### 2.5. Construction of shRNA Silencing Cells

The CDS region sequence of pig *GLP-2* (NM_214324.1) gene was obtained from the gene sequence published in the GenBank database. Three target sequences (shRNA1, shRNA2, shRNA3) through the shRNA target gene design website (http://rnaidesigner.thermofisher.com/rnaiexpress/, 15 July 2020) were screened. Primers were synthesized by Sangon Biotech Co., Ltd. (Shanghai, China). Detailed information is shown in Table 2. The three pairs of shRNA sequences were oligo annealed and connected. The oligo annealing procedure and conditions were as follows: 5 μL each for positive and negative strands, 2 μL annealing buffer (10×), 8 μL ddH_2_O and then 95 °C 5 min, natural cooling at room temperature for 20 min. The connection procedure and conditions were as follows: 4 μL of oligo product, 2 μL of linearization vector, 1 μL of T4 DNA ligase, 4 μL of ligation buffer (5×), 9 μL of ddH_2_O and then 16 °C overnight connection. After transforming the vector into DH5α, we performed plasmid extraction. According to the instruction of lipofectamine 2000 (Invitrogen, Grand Island, NY, USA) transfection reagent, we transfected the transfection reagent, plasmid and Opti-MEM (Gibco, Grand Island, NY, USA) into the 12-well plate. After 48 h of drug screening, we observed the fluorescence. When the cell confluency was greater than 80%, the cells were collected and total RNA was extracted to verify the interference efficiency.

### 2.6. PEDV Infects IPEC-J2 Cells and shRNA Silencing Cells

The *GLP-2* gene interfering IPEC-J2 cells and control IPEC-J2 cells were seeded in 6-well plates at 1 × 10^5^. They were cultured in Dulbecco’s modified Eagle medium (DMEM; Gibco, Grand Island, NY, USA) containing 10% fetal bovine serum (FBS; Gibco, Grand Island, NY, USA) at 5%CO_2_ and 37 °C and then cultured for 12 h. PEDV was added to the experimental wells at MOI = 0.1, and cells without virus infection were set as mock cells. After incubation for 48 h, the cells were collected and used for total RNA isolation.

### 2.7. Total RNA Extraction and qPCR

The Trizol method was used to extract total RNA from piglets’ intestinal tissues and IPEC-J2 cells. The extraction steps were strictly in accordance with the Trizol Reagent (Takara, Beijing, China). The purity of RNA was detected by 2.2% formaldehyde denaturation agarose gel electrophoresis and NanoDrop ND-1000 micro nucleic acid concentration analyzer. RNA was anti-transcribed into cDNA and used as template. We used qPCR to quantify gene expression, with a volume of 20 μL containing cDNA template 2.0 μL, forward and reserve primers each 0.4 μL (10 μmol/L), ROX Reference Dye II (50×) 0.4 μL, SYBR Green qPCR Master Mix (2×) 10 μL, ddH2O 6.8 μL. The reaction conditions were as follows: 95 °C for 5 min, followed by 40 cycles of 95 °C for 10 s and 60 °C for 30 s. We analyzed the melting curve after the amplification was completed and judged the unity of PCR amplification according to whether there was an 85 ± 0.8 °C peak on the melting curve.

### 2.8. Western Blotting Analysis

Cells were washed twice with PBS and incubated on ice with RIPA Lysis and Extraction Buffer (biosharp, BL509A, wobio, Nanjing, China) and Protease Inhibitor Cocktail (biosharp, BL612A, wobio, Nanjing, China). The lysates were denatured for 10 min in 5×SDS-PAGE loading buffer and separated with SDS-PAGE. The proteins were then transferred to nitrocellulose Western blotting membranes (millipore, IPVH08100, wobio, Nanjing, China). The membranes were blocked with PBS containing 5% nonfat dry milk and 0.2% Tween 20 for 2 h at room temperature. Then, the membranes with the primary antibody were incubated at 4 °C for the whole night. After the membranes were washed with PBS containing 0.1% Tween 20 (0.1% PBST), they were incubated with the corresponding secondary antibody (conjugated with HRP) at room temperature for 2 h and detected with enhanced chemiluminescence (ECL) (Thermo Fisher Scientific, 34580, Waltham, MA, USA).

### 2.9. Statistical Analysis

According to the PEDV amplification standard curve y = −3.3354lg(x) + 37.832 (y represents the number of Ct cycles, x represents the logarithm of the virus copy number based on the base 10), the quantitative Ct value of the PEDV M gene was substituted into the formula and then the number of copies of PEDV was calculated. The 2^−ΔΔCt^ method was used to detect the transcript levels of the genes. ΔΔCt = (the average Ct value of the target gene in the test group − the average Ct value of the internal reference gene in the test group) − (the average Ct value of the target gene in the control group − the average Ct value of the internal reference gene in the control group). SPSS 21.0 software was utilized to carry out an independent sample t-test, and data are expressed as means ± standard deviation (Mean ± SD).

## 3. Results

### 3.1. Hematoxylin and Eosin (H&E) Staining of the Small Intestine

In order to detect the difference in intestinal structures between PEDV-infected piglets and healthy piglets, the intestinal mucosa of piglets was stained by H&E staining. The results showed that the intestinal villi of the diseased piglets were severely damaged, the intestinal mucosal epithelial cells were necrotizing and shedding, the lamina propria were exposed and the intestinal glands were atrophied. In healthy piglets, the structure of the intestinal mucosa was complete, with distinct layers, and the intestinal mucosal epithelial cells had a clear outline and regular arrangement (Figure 1). The above results indicated that the intestines exhibited obvious lesions, and the intestinal mucosa was damaged post-PEDV infection. The intestines derived from PEDV-infected and healthy piglets were used for subsequent experiments

### 3.2. Total RNA Quality Detection and qPCR Primer Specificity

The total RNA extracted from the intestinal tissue and IPEC-J2 were subjected to 2.2% formaldehyde denaturation agarose gel electrophoresis, and the three bands of 28S, 18S and 5S were clear (Appendix A). This proved that there was no DNA contamination band and degradation. The A260/A280 of the samples were 1.8–2.0, indicating that the completeness and purity of the RNA extraction were good, and it could be used for subsequent experiments. The qPCR amplification and melting curves for the *GLP-2* and PEDV M genes showed that the PCR product had only one definite peak and no primer dimer or non-specific product (Appendix A)

### 3.3. Expression Analysis of GLP-2 Gene in Intestinal Segments of the Healthy and Damaged Intestinal Mucosa Group

Gene expression quantitative results found that the *GLP-2* gene was expressed in the duodenum, jejunum and ileum. In addition, the expression level of *GLP-2* gene in the jejunum and ileum of the intact intestinal mucosa group was significantly higher than that in the intestinal mucosal damage group (*p* < 0.01) (Figure 2). The results showed that when piglets were infected with PEDV, the expression of the *GLP-2* gene in jejunum and ileum of piglets decreased significantly.

### 3.4. Analysis of Differential Expression of GLP-2 Gene in IPEC-J2 Cells with PEDV Infection

To further explore the relationship between *GLP-2* gene expression and PEDV infection, we used qPCR to detect the difference between the expression level of *GLP-2* gene in IPEC-J2 cells infected with PEDV for 48 h and uninfected control cells. The results showed that the expression level of *GLP-2* gene in PEDV-infected cells was significantly lower than that in the control group (*p* < 0.01) (Figure 3). 

### 3.5. Verification of GLP-2 Gene Interference Efficiency in IPEC-J2 Cells

To investigate the potential role of *GLP-2* gene in regulating PEDV-infection, we constructed *GLP-2* gene-silencing cells. After transfection of the *GLP-2* gene shRNAs into IPEC-J2 cells, the red fluorescent protein could be expressed, indicating that shRNAs had been successfully transfected into cells and stably expressed (Figure 4A). The qPCR assay revealed that the interference efficiency of shRNA1 and shRNA2 reached 61%, which could be used for subsequent experiments (Figure 4B). The above results indicated that the *GLP-2* gene silencing cells had been successfully constructed.

### 3.6. Effect of GLP-2 Gene Silencing on PEDV Infection

To further verify the effects of *GLP-2* gene expression on PEDV infection, total RNA from shRNA1 and shRNA2 interfering cells infected by PEDV was extracted for M gene copy number detection. The results showed that the copy number of the PEDV M gene in the GLP-2 interfering cells was significantly higher than that in the control group (*p* < 0.01) (Figure 5A). Western blot analysis further validated the increased PEDV N protein expression in the GLP-2 interfering cells (Figure 5B). The results show that when the *GLP-2* gene expression was down-regulated, the PEDV copy number was increased, indicating that *GLP-2* gene may be involved in regulating the proliferation of PEDV in host cells.

## 4. Discussion

Piglet diarrhea is caused by many factors, of which PEDV-induced porcine epidemic diarrhea is the most common viral diarrhea disease in piglets, causing serious economic losses to the pig industry [23]. The small intestine is an important immune organ that resists the invasion of foreign pathogens through the intestinal epithelial villi. Therefore, maintaining the integrity of the intestinal mucosa is essential for animal health. We observed a damaged intestinal mucosa integrity induced by PEDV infection. It has been reported that GLP-2 is an intestinal protective factor that widely participates in the repair of intestinal mucosa [24]. Our study found that the *GLP-2* gene is expressed in all intestinal segments, indicating that the *GLP-2* gene may hae physiological functions in the intestinal mucosal immune responses. Expression of the *GLP-2* gene in the jejunum and ileum of the intestinal mucosal damaged group was lower than that of the control group, indicating that PEDV infection may affect the integrity of intestinal mucosa by regulating the expression of the *GPL-2* gene. 

Studies have shown that GLP-2 receptor (GLP-2R) produces a large amount of cAMP by activating the activity of adenylate cyclase. GLP-2 can accelerate the differentiation, growth and repair of the entire intestinal crypt cell population through the cAMP-dependent signal transduction pathway mediated by GLP-2R [25]. GLP-2 can also promote the recovery of pathological intestinal mucosa and induce the proliferation of intestinal epithelial cells through the cell signal transduction pathway of mitogen-activated protein kinases [26]. Moreover, GLP-2 regulates the gene expression of small intestinal tight junction proteins. Tight junction proteins are an important connection method of intestinal epithelial cells and play an important role in maintaining the integrity of the intestinal barrier. After PEDV infection, the expression levels of tight junction proteins such as ZO-1, ZO-2, Occludin and Claudin-4 in the intestines significantly changed [22]. In addition, GLP-2 can stimulate the proliferation of small intestinal fossa cells to inhibit cell apoptosis, compensate for intestinal development disorders, and reduce enteritis symptoms in newborn animals [27,28]. We further detected the changes in *GLP-2* gene expression at the cellular level and found expression of *GLP-2* gene at 48 h after PEDV infection was significantly decreased, indicating that *GLP-2* gene may be involved in the repair of intestinal mucosal damage induced by PEDV infection. 

The most common way for a virus to infect the host is to destroy the epithelial barrier of the host cell, and tight junctions are crucial to maintaining the cell barrier [29]. Jung et al. [30] proved that PEDV infection caused changes in the structure of tight junction proteins ZO-1 and E-cadherin in the villi epithelium in vivo. Other viruses that cause diarrhea, such as rotavirus, can infect intestinal epithelial cells and the distribution of tight junction proteins Claudin-1 and Occludin, thereby promoting viral infections [31]. To further confirm the relationship between *GLP-2* gene and PEDV, we silenced the expression of *GLP-2* gene and observed the increased PEDV copies in host cells upon PEDV infection, which provided further supports for the participation of the *GLP-2* gene in mediating the interactions between PEDV and host cells. The mechanisms of interaction between the *GLP-2* gene and PEDV infection warrant further investigations. 

## 5. Conclusions

In conclusion, we found that PEDV infection causes obvious pathological changes in the porcine intestines and that *GLP-2* gene is involved in regulating PEDV infection. Furthermore, higher expression of *GLP-2* gene may be beneficial to resist PEDV infection. Our findings provide novel insights into the functions of *GLP-2* gene in response to PEDV infection and may contribute to the identification of resistant genes or genetic markers for disease resistance breeding in pigs. 

## Figures and Tables

**Figure 1 animals-11-00644-f001:**
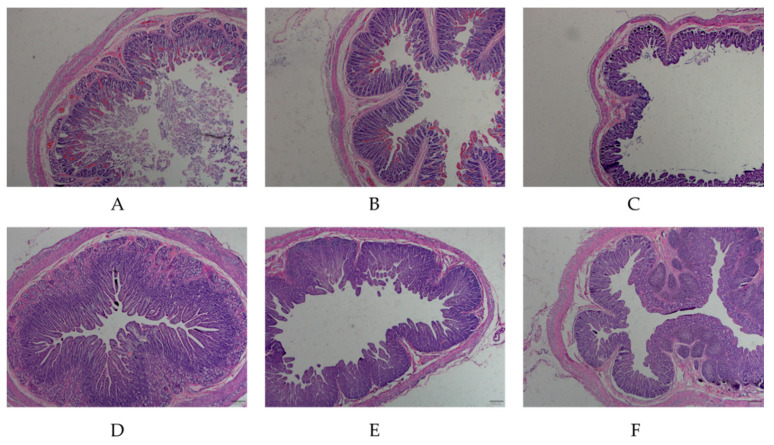
H&E-staining microscopic examination of the small intestine tissue from porcine epidemic diarrhea virus (PEDV)-infected and normal piglets. (**A**–**C**) indicate the duodenum, jejunum, ileum tissues of PEDV-infected piglets (40×); (**D**–**F**) indicate the duodenum, jejunum, ileum tissues of normal piglets (40×).

**Figure 2 animals-11-00644-f002:**
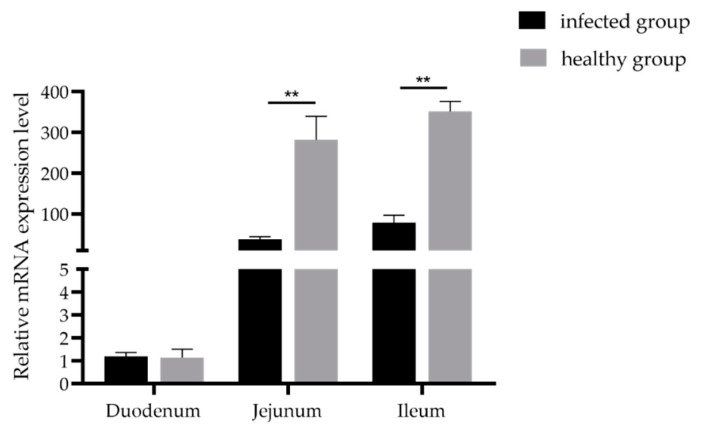
Expression levels of the *GLP-2* gene in the PEDV-infected group and healthy group. ** *p* < 0.01. Bars represent the mean ± SD (*n* = 3).

**Figure 3 animals-11-00644-f003:**
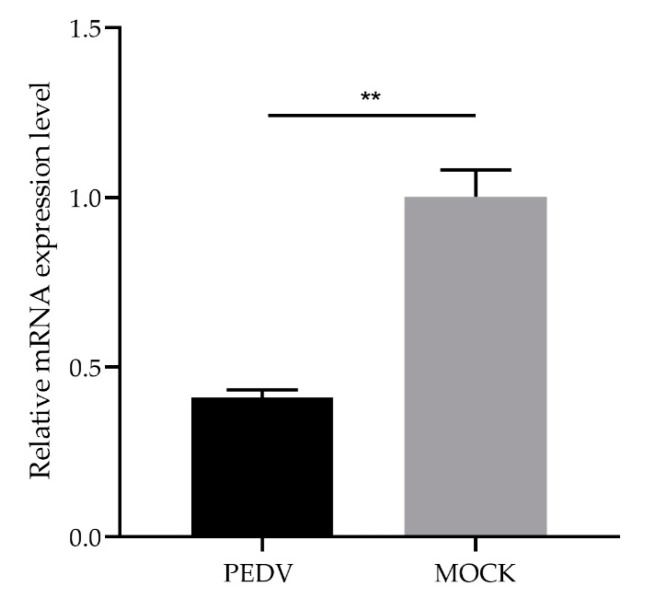
Expression changes of the *GLP-2* gene in PEDV-infected IPEC-J2 cells. ** *p* < 0.01. Bars represent the mean ± SD (*n* = 3).

**Figure 4 animals-11-00644-f004:**
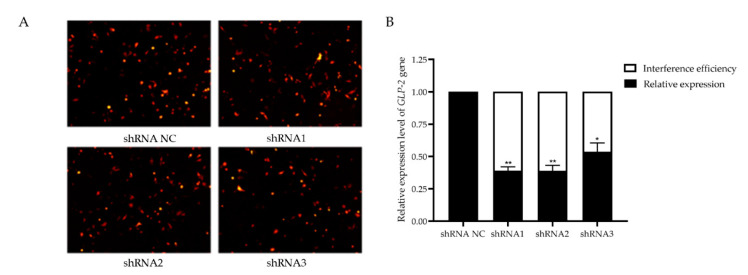
Verification of *GLP-2* gene interference efficiency. (**A**) Transfection of *GLP-2* gene shRNAs vector into IPEC-J2 cells. (**B**) *GLP-2* gene mRNA expression level. shRNA NC represents negative control. shRNA1 shRNA2 and shRNA3 represent three RNA interference fragments of the *GLP-2* gene. * *p* < 0.05, ** *p* < 0.01. Bars represent the mean ± SD (*n* = 3).

**Figure 5 animals-11-00644-f005:**
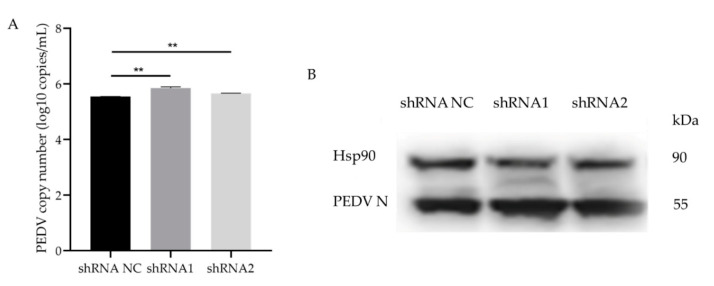
PEDV infects *GLP-2* gene silent cell lines. (**A**) qPCR for determination of PEDV copy number in shRNA NC, shRNA1 and shRNA2 cells. (**B**) Western blot analysis of PEDV N protein. shRNA NC represents negative control. shRNA1 shRNA2 and shRNA3 represent three RNA interference fragments of *GLP-2* gene. ** *p* < 0.01. Bars represent the mean ± SD (*n* = 3).

**Table 1 animals-11-00644-t001:** The primer sequence of genes for qPCR.

Gene	Primer Sequence	Product Length (bp)
*GLP-2*	F: 5′-ACTCACAGGGCACGTTTACCA-3′ R: 5′-AGGTCCCTTCAGCATGTCTCT-3′	150
*PEDV M*	F: 5′-AGGTCTGCATTCCAGTGCTT-3′ R: 5′-CCTGCCCAGATTCAGCAAAG-3′	216
*GAPDH*	F: 5′-ACATCATCCCTGCTTCTACTGG-3′ R: 5′-CTCGGACGCCTGCTTCAC-3′	187
*β-ACTIN*	F: 5′-TGGCGCCCAGCACGATGAAG-3′ R: 5′-GATGGAGGGGCCGGACTCGT-3′	149

**Table 2 animals-11-00644-t002:** The oligomeric single-stranded DNA sequence of shRNAs.

Name	Sequence of Oligo
*shRNA1*	F: 5′-CACCG$\underline{\mathrm{CCCTCTCGATGATCCAGATT}\underline{\mathrm{TCAAGAG}}\mathrm{AATCTGGATCATCGAGAGGG}}$TTTTTTG -3′ R: 5′-*GATCC*AAAAAA$\underline{\mathrm{CCCTCTCGATGATCCAGATT}\underline{\mathrm{CTCTTGA}}\mathrm{AATCTGGATCATCGAGAGGG}}$C-3′
*shRNA2*	F: 5′-CACC$\underline{\mathrm{GCCACCCGAGACTTTATAAT}\underline{\mathrm{TCAAGAG}}\mathrm{ATTATAAAGTCTCGGGTGGC}}$TTTTTTG-3′ R: 5′-*GATCC*AAAAAA$\underline{\mathrm{GCCACCCGAGACTTTATAAT}\underline{\mathrm{CTCTTGA}}\mathrm{ATTATAAAGTCTCGGGTGGC}}$-3′
*shRNA3*	F: 5′-CACCG$\underline{\mathrm{CTTTGTGGCTGGATTGTT}\underline{\mathrm{TCAAGAG}}\mathrm{AACAATCCAGCCACAAAG}}$TTTTTTG-3′ R: 5′-*GATCC*AAAAAA$\underline{\mathrm{CTTTGTGGCTGGATTGTT}\underline{\mathrm{CTCTTGA}}\mathrm{AACAATCCAGCCACAAAG}}$C-3′
*shRNA NC*	F: 5′-CACCG$\underline{\mathrm{TTCTCCGAACGTGTCACGT}\underline{\mathrm{CAAGAGATT}}\mathrm{ACGTGACACGTTCGGAGAA}}$TTTTTTG-3′ R: 5′- *GATCC*AAAAAA$\underline{\mathrm{TTCTCCGAACGTGTCACGT}\underline{\mathrm{AATCTCTTG}}\mathrm{ACGTGACACGTTCGGAGAA}}$C-3′

The italic part is the enzyme cutting site; the single-underlined part is the interference sequence and its complementary sequence, and the double-underlined part is the loop sequence.

## Data Availability

The data presented in this study are available on reasonable request from the corresponding author.

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
