# Peer review of "Analysis of Intestinal Mucosa Integrity and GLP-2 Gene Functions upon Porcine Epidemic Diarrhea Virus Infection in Pigs"

_animals, 2021, doi:10.3390/ani11030644_

Round 1
Reviewer 1 Report
Dear authors,
many thanks for that very interesting paper and your clear investigations. I have ranked your work into „minor revision“ because I have only a few questions:
L100: How did you check, that piglets were infected with PEDV.
L103: What do you mean by „family screening“? This can only be understood be reading [21] but might be important for understanding.
L136: I have to admit, that I am not an expert in that field. However, is it normal, that fluorescence appears after 48 h and at which time became fluorescence full.
L145: Similar to L136, I thing that it is useful to comment on the time of incubation of 48 hours. Is it sufficient?
L170: Where does the PEDV amplification standard curve comes from?
L193: Figure 1: 3 out of 6 samples per group were used in Fig. 1. Are these sample representative for the samples not shown?
L230: Do you have any explanation why shRNA3 was less efficient than shRNA1 or 2. It looks like, that you don‘t trust the efficiency of shRNA3.
L243: Regarding Fig. 5B, I have problems to retrace an increased PEDV N protein expression in the GLP-2 interfering cells.
L248: There are amazingly low SE. How many independent sample did you use in your experiment. I would prefer, that you will provide this information for all parts of you experiment.
Author Response
Dear editor,
Thank you for your careful revision and comments concerning our manuscript. We have made correction which we hope meet with approval. Revised portion are marked in different color in the paper. The main corrections in the paper and the responds to reviewer’s questions are as flowing:
L100: How did you check, that piglets were infected with PEDV.
Response: Intestinal contents was used to check PEDV by qPCR, and the products were examined by agarose gel electrophoresis. If the PEDV genome was detected, the piglets were considered infected with PEDV. We have provided the method in our manuscript (lines 97-98 on page 3). The detailed method had been reported in our previous report (Zong et al., 2019).
Zong, Q.F.; Huang, Y.J.; Wu, L.S.; Wu, Z.C.; Wu, S.L.; Bao, W.B. Effects of porcine epidemic diarrhea virus infection on tight junction protein gene expression and morphology of the intestinal mucosa in pigs. Pol J Vet Sci 2019, 345-353.
L103: What do you mean by “family screening”? This can only be understood be reading [21] but might be important for understanding.
Response: The six piglets infected with PEDV and six control piglets were half-sibs. We have modified the statements in our revision (line 98 on page 3).
L136: I have to admit, that I am not an expert in that field. However, is it normal, that fluorescence appears after 48 h and at which time became fluorescence full.
Response: Thanks for your suggestions. We have improved our presentation in our revisions. When the cell confluency greater than 80%, the cells were collected (lines 131-132 on page 3).
L145: Similar to L136, I thing that it is useful to comment on the time of incubation of 48 hours. Is it sufficient?
Response: Incubation of 48 h is sufficient and widely used in studies on PEDV infections (Wang et al., 2020; Xue et al., 2017; Zhao et al.,2014).
Wang, S.Q.; Wu, J.Y.; Wang, F.; Wang, H.F.; Wu, Z.C.; Wu, S.L.; Bao W.B. Expression pattern analysis of antiviral genes and inflammatory cytokines in PEDV-infected porcine intestinal epithelial cells. Front Vet Sci 2020.
Xue, M.; Zhao, J.; Ying, L.; Fu, F.; Li, L.; Ma, Y.L.; Shi, H.Y.; Zhang, J.E.; Feng, L.; Liu, P.H. IL-22 suppresses the infection of porcine enteric coronaviruses and rotavirus by activating STAT3 signal pathway. Antivir Res 2017, 68-75.
Zhao, S.S.; Gao, J.K.; Zhu, L.K.; Yang, Q. Transmissible gastroenteritis virus and porcine epidemic diarrhoea virus infection induces dramatic changes in the tight junctions and microfilaments of polarized IPEC-J2 cells. Virus Res 2014, 34-45.
L170: Where does the PEDV amplification standard curve comes from?
Response: The standard curve equation was previously established by our lab. Based on the PEDV M gene sequence, we designed qPCR amplification primers. The qPCR product of M gene was ligated into the pMD18T plasmid to construct a recombinant plasmid. The recombinant plasmid was treated with ten-fold serial dilutions and each dilution was amplified by qPCR. The standard curve presented in our manuscript was obtained by the qPCR Ct values and plasmid concentrations. In this this, we quantified the PEDV copies using qPCR, and the Ct value was substituted into the standard curve to calculate the virus copies.
L193: Figure 1: 3 out of 6 samples per group were used in Fig. 1. Are these samples representative for the samples not shown?
Response: All the intestinal samples from infected piglets showed damaged intestinal villi, and those from healthy piglets showed normal intestinal villi. The representative samples from the infected and healthy piglets were shown in Fig. 1.
L230: Do you have any explanation why shRNA3 was less efficient than shRNA1 or 2. It looks like, that you don’t trust the efficiency of shRNA3.
Response: The interference efficiency of shRNA1 and shRNA2 were more than 60%, the interference efficiency of shRNA3 was lower than 50%. To get better interference effects on GLP-2 expression, we chose shRNA1 and shRNA2 for subsequent experiments.
L243: Regarding Fig. 5B, I have problems to retrace an increased PEDV N protein expression in the GLP-2 interfering cells.
Response: We had analyzed the gray values ​​of the bands through the image J software. The value of shRNA1 (1.74) and shRNA2 (1.53) were higher than that of shRNA NC (1.32). The results indicated an increased PEDV N protein expression.
L248: There are amazingly low SE. How many independent sample did you use in your experiment. I would prefer, that you will provide this information for all parts of you experiment.
Response: The viral copies have been transformed by log10, so the standard deviation (SD) was relatively low. Thee independent samples were used in the experiment. The information for other part has been provided in our revised manuscript (line 212 on page 6; line 221 on page 6; line 233 on page 7; lines 246-247 on page 7).

Reviewer 2 Report
The paper „Analysis of intestinal mucosa integrity and GLP-2 gene functions upon Porcine Epidemic Diarrhea Virus Infection in Pigs” by Zhou et al. presents results of the study designed to test the hypothesis that the PEDV infection leads to disrupted intestinal integrity and is mediated by the GLP-2 activity. The in vivo and in vitro experiments carried out to verify this hypothesis were scientifically sound. The GLP-2 activity was confirmed on mRNA and protein level. Additionally, RNA interference was used to create dedicated model in vitro. The paper is well drafted, only few comments regarding this study:
Introduction:
- Page 2, Line 54: The Authors omitted a basic function of the intestine and the GALT, which is a local immune system, please add “Intestine is an important part of the local and the systemic immune system”. Please add reference, for example this: https://www.ncbi.nlm.nih.gov/pmc/articles/PMC3292882/
- Page 2, Lines: 85-92: Maybe the Authors could remove abstract of their materials and methods, results and discussion from the Introduction and simply formulated the hypotheses and the research goals?
Materials and Methods:
- The style of this chapter is inconsistent. The beginning of the chapter is written in the passive voice, e.g., “the experimental pigs were selected from…”, then it turns to active voice, e.g., “Then we rinsed them with PBS…..”
- Even though I understand that active voice add dynamics to the narrative, the materials and methods are still best drafted in the passive voice. Please review the “Materials and methods” like that.
- Provide culture conditions for IPEC-J2
Results:
- Page 5, Line 184-185: necrosis and atrophy are nouns not verbs, please review those sentences accordingly
- Page 6, Line 220: It is enough to say that something is significantly lower, “extremely significantly lower” is already too much
- Page 6, Lines 221-223: These comments belong to a discussion, remove from this section
Minor comments:
Line 105: on-site
Line 109: fixed
Line 151: replace “reserved” with another verb
Line 189: post-PEDV
Author Response
Dear editor,
Thank you for your careful revision and comments concerning our manuscript. We have made correction which we hope meet with approval. Revised portion are marked in different color in the paper. The main corrections in the paper and the responds to reviewer’s questions are as flowing:
Introduction:
- Page 2, Line 54: The Authors omitted a basic function of the intestine and the GALT, which is a local immune system, please add “Intestine is an important part of the local and the systemic immune system”. Please add reference, for example this: https://www.ncbi.nlm.nih.gov/pmc/articles/PMC3292882/
Response:Thanks for your suggestions. The reference has been cited in our revised manuscript (lines 324-325 on page 9).
- Page 2, Lines: 85-92: Maybe the Authors could remove abstract of their materials and methods, results and discussion from the Introduction and simply formulated the hypotheses and the research goals?
Response:Thanks for your suggestions. We have revised this section according to your suggestions (lines 82-88 on page 2).
Materials and Methods:
- The style of this chapter is inconsistent. The beginning of the chapter is written in the passive voice, e.g., “the experimental pigs were selected from…”, then it turns to active voice, e.g., “Then we rinsed them with PBS…..”
Response:Thanks for your suggestions. We have revised our manuscript (lines 97-101 on page 3).
- Even though I understand that active voice add dynamics to the narrative, the materials and methods are still best drafted in the passive voice. Please review the “Materials and methods” like that.
Response:Thanks for your comments. The “Materials and methods” have been drafted in the passive voice.
- Provide culture conditions for IPEC-J2
Response:Thanks for your comments. We have provided culture conditions for IPEC-J2 (lines 138-140 on page 4)
Results:
- Page 5, Line 184-185: necrosis and atrophy are nouns not verbs, please review those sentences accordingly
Response: Done.
- Page 6, Line 220: It is enough to say that something is significantly lower, “extremely significantly lower” is already too much
Response: Done.
- Page 6, Lines 221-223: These comments belong to a discussion, remove from this section
Response: Done.
Minor comments:
Line 105: on-site
Response: Done.
Line 109: fixed
Response: Done.
Line 151: replace “reserved” with another verb
Response: Done.
Line 189: post-PEDV
Response: Done.
